# Faster rehabilitation weight gain during childhood is associated with risk of non-communicable disease in adult survivors of severe acute malnutrition

**Debbie S. Thompson**[1]*, **Kimberley McKenzie**[1], **Charles Opondo**[2], **Michael S. Boyne**[3], **Natasha Lelijveld**[4,5], **Jonathan C. Wells**[6], **Tim J. Cole**[6], **Kenneth Anujuo**[4], **Mubarek Abera**[7], **Melkamu Berhane**[7], **Albert Koulman**[8], **Stephen A. Wootton**[9], **Marko Kerac**[4,5]*, **Asha Badaloo**[1], **CHANGE Study Collaborators Group**[¶]

1 Caribbean Institute for Health Research, The University of the West Indies, Kingston, Jamaica, 2 Department of Medical Statistics, Faculty of Epidemiology & Population Health, London School of Hygiene & Tropical Medicine, London, United Kingdom, 3 Department of Medicine, The University of the West Indies, Kingston, Jamaica, 4 Department of Population Health, Faculty of Epidemiology and Population Health, London School of Hygiene and Tropical Medicine, London, United Kingdom, 5 Centre for Maternal, Adolescent & Reproductive Child Health (MARCH), London School of Hygiene & Tropical Medicine, London, United Kingdom, 6 Population Policy and Practice Department, UCL Great Ormond Street Institute of Child Health, London, United Kingdom, 7 Faculty of Medical Science, Institute of Health, Jimma University, Jimma, Ethiopia, 8 Nutritional Biomarker Laboratory, MRC Epidemiology Unit, University of Cambridge, Cambridge, United Kingdom, 9 Southampton NIHR Biomedical Research Centre, University of Southampton, Southampton, United Kingdom

¶ Membership of the CHANGE Study Collaborators Group is provided in the Acknowledgments.
* debbie.thompson@uwimona.edu.jm (DST); Marko.Kerac@lshtm.ac.uk (MK)

**Data Availability Statement:** Data for this manuscript can be accessed at London School of

## Abstract

Nutritional rehabilitation during severe acute malnutrition (SAM) aims to quickly restore body size and minimize poor short-term outcomes. We hypothesized that faster weight gain during treatment is associated with greater cardiometabolic risk in adult life. Anthropometry, body composition (DEXA), blood pressure, blood glucose, insulin and lipids were measured in a cohort of adults who were hospitalized as children for SAM between 1963 and 1993. Weight and height measured during hospitalization and at one year post-recovery were abstracted from hospital records. Childhood weight gain during nutritional rehabilitation and weight and height gain one year post-recovery were analysed as continuous variables, quintiles and latent classes in age, sex and minimum weight-for-age z-scores-adjusted regression models against adult measurements. Data for 278 adult SAM survivors who had childhood admission records were analysed. Of these adults, 85 also had data collected 1 year post-hospitalisation. Sixty percent of participants were male, mean (SD) age was 28.2 (7.7) years, mean (SD) BMI was 23.6 (5.2) kg/m². Mean admission age for SAM was 10.9 months (range 0.3–36.3 months), 77% were wasted (weight-for-height z-scores<-2). Mean rehabilitation weight gain (SD) was 10.1 (3.8) g/kg/day and 61.6 (25.3) g/day. Rehabilitation weight gain > 12.9 g/kg/day was associated with higher adult BMI (difference = 0.5 kg/m², 95% CI: 0.1–0.9, *p* = 0.02), waist circumference (difference = 1.4 cm, 95% CI: 0.4–2.4, *p* = 0.005), fat mass (difference = 1.1 kg, 95% CI: 0.2–2, *p* = 0.02), fat mass index (difference =

Hygiene and Tropical Medicine Data Compass Repository https://datacompass.lshtm.ac.uk/id/eprint/2656/. The link to the wider 'parent' project is https://datacompass.lshtm.ac.uk/id/eprint/2655/

**Funding:** This study was funded by the UK Medical Research Council (MRC) / Global Challenges Research Fund (GCRF), grant number: MR/V000802/1, received by MK. The original project was supported by the New Zealand Health Research Council Grant 09/052, Developmental Adaptation to an Obesogenic Environment Program. AK was also supported by the NIHR Cambridge Biomedical Research Centre (NIHR203312). The views expressed are those of the authors and not necessarily those of the NIHR or the Department of Health and Social Care.

**Competing interests:** The authors have declared that no competing interests exist.

0.32kg/m$^2$, 95% CI: -0.0001–0.6, $p = 0.05$), and android fat mass (difference = 0.09 kg, 95% CI: 0.01–0.2, $p = 0.03$). Post-recovery weight gain (g/kg/month) was associated with lean mass (difference = 1.3 kg, 95% CI: 0.3–2.4, $p = 0.015$) and inversely associated with android-gynoid fat ratio (difference = -0.03, 95% CI: -0.07 to -0.001 $p = 0.045$). Rehabilitation weight gain exceeding 13g/kg/day was associated with adult adiposity in young, normal-weight adult SAM survivors. This challenges existing guidelines for treating malnutrition and warrants further studies aiming at optimising these targets.

## Introduction

Severe acute malnutrition (SAM) remains an important cause of mortality globally, and malnutrition contributes to the deaths of more than 500,000 children under the age of 5 years annually [1]. As global efforts improve the management of SAM, and with the availability of ready to use therapeutic foods, reports suggest that the mortality from SAM is declining in settings such as the Bale region of Ethiopia [2] and in Zambia [3]. However, while the short-term outcomes of severe childhood malnutrition may be improving, the long-term consequences are less clear. Although there is evidence of an association between prenatal undernutrition and higher risk of chronic non-communicable diseases (NCDs) at the later stages of life [4, 5], much of this evidence is from developed countries. Evidence of links between infant and child malnutrition and later NCDs is more limited [6–9], and potential mechanisms are yet to be identified [10].

Most malnutrition treatment programs rely on the attainment of an acceptable weight-for-height as an indicator of recovery. To this end, energy-dense feeds are usually provided during the rehabilitation stage of treatment of severe malnutrition, during which children are fed increasing amounts of energy and protein-enriched feeds to achieve a normal body weight. The recommended energy intake during the transition period is 100–135 kcal/kg/day [11], followed by intake of 150–225 kcal/kg/d during the phase when catch-up weight gain occurs, with the goal of gaining $\geq$ 10 g/kg/day as recommended by the World Health Organization (WHO) [12]. However, it is unclear whether better short-term outcomes occur at the expense of longer-term outcomes. The thrifty phenotype hypothesis proposes that poor foetal/ infant growth is a risk factor for the subsequent development of type 2 diabetes and the metabolic syndrome due to the effects of poor early-life nutrition [13]. In such "metabolically thrifty" individuals, faster weight gain in the plastic developmental period of infancy could pose an additional risk for later NCDs [14].

Many countries with high levels of early childhood malnutrition also have high burdens of obesity and NCDs including hypertension, type 2 diabetes and cardiovascular disease [15]. Data from the WHO indicate that NCDs are the leading causes of mortality and disability globally, causing the deaths of more than three in five people worldwide and responsible for more than half the global burden of disease [16]. NCDs also cause and perpetuate poverty while hindering economic development in low- and middle-income countries (LMICs) as they struggle with the economic burden of providing treatment for individuals with NCDs, thus recapitulating the cycle.

This study aims to explore patterns of rehabilitation and post-recovery weight gain in children who were hospitalized with SAM, and to investigate the associations between rehabilitation and post-recovery weight gain and adult obesity, body composition and NCD risk. We

hypothesize that faster rates of weight gain during nutritional rehabilitation are associated with later NCD risk.

## Methods

Setting: The LION (Long-term Implications of Nutrition) Cohort was retrospectively assembled from individuals who were admitted to the Tropical Metabolism Research Unit (TMRU) ward at the University Hospital of the West Indies between the years 1963–1993 with a diagnosis of SAM. Members of the cohort were followed up and extensively characterised at their adult stage between 2008–2012 in the Jamaica Marasmus and Kwashiorkor Adult Survivors (JAMAKAS) Study (**Fig 1**) [6, 17–20].

This study is a secondary analysis of data collected from the adults who participated in the JAMAKAS Study for whom childhood hospital admission records were available. The Mona Campus Research Ethics Committee of the University of the West Indies (UWI) approved the original follow-up study (ECP/71, 2008/2009) and all participants provided written informed consent. Written informed consent was obtained from the parent/guardian of each participant under 18 years of age. Ethical approval for this secondary analysis was obtained from both UWI (CREC-MN. 204 20/21) and LSHTM Research and Ethics committee (ref 27436). Patients or the public were not involved in the design, or conduct, or reporting, or dissemination plans of our research.

During hospitalization for SAM, treatment began with an acute "stabilisation" phase during which infections and fluid imbalance were treated and a standardised diet was offered to provide energy for weight maintenance. With clinical improvement, resolution of oedema and improved appetite and affect, the "rehabilitation" phase occurred, during which children were fed increasing amounts of energy and protein-enriched feeds to attain a healthy body weight (90–110% weight-for-height, using the NCHS standard). The final phase of treatment, referred to as "recovery", involved weaning the child to an age-appropriate diet before discharge from hospital. Children were discharged having attained weight-for-height ≥ 90% using the National Center for Health Statistics (NCHS) growth standards [21], and were followed up as outpatients for up to two years post-hospitalisation. During this period, they were fed a home-based diet.

Participant Characteristics: Childhood data abstracted from hospital records included birth weight, admission weight and height/length and weight during hospitalization and one year post-discharge from hospital. Adult data were collected at the time of the JAMAKAS Study (2008–2012) and included a detailed medical and drug history, measurements of anthropometry, blood pressure, body composition (by dual-energy x-ray absorptiometry, GE Lunar Prodigy), fasting glucose, insulin and serum lipids. These archived childhood and adult data were accessed from medical records between September 1, 2021, and July 31, 2022. Authors had access to information that could identify individual participants during, but not after, data collection.

Exposure variables: We derived six measures of post-malnutrition growth (i.e., weight and height gain) based on available data and established assessment methods (**Table 1**). The first three measures were derived from in-patient weight and age data starting at the time of minimum weight, which was usually taken at admission or after oedema had resolved and continuing until the time of maximum weight during treatment. These measures respectively utilized the daily change in weight-for-age z-score (WAZ) (using WHO 2006 growth standards) [22], grams/kilogram (g/kg) of body weight and weight in grams (**Table 1**), all common measures of growth in malnutrition programs. Height-for-age z-score (HAZ) was not used during this period as height gain was small and HAZ is usually an indicator for chronic malnutrition but

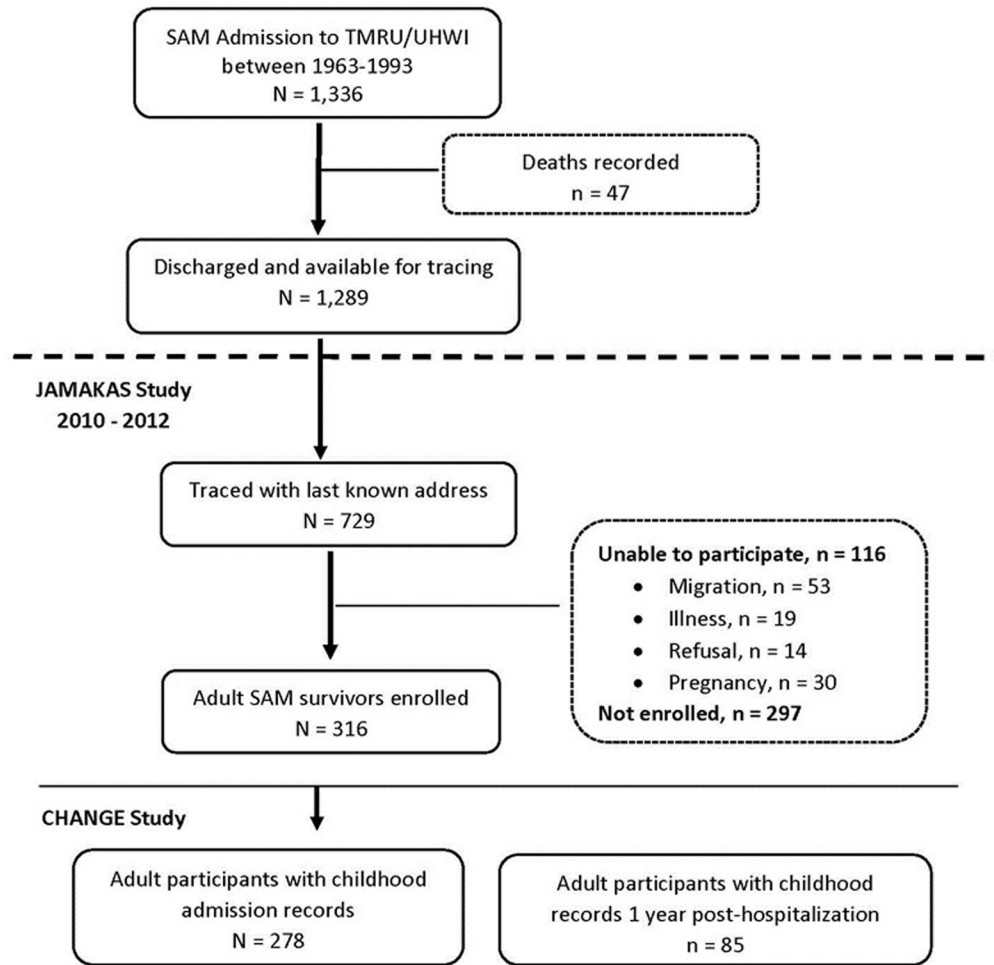

**Fig 1. Flow chart detailing recruitment of adult survivors of severe acute malnutrition (SAM) (*n* = 278).** "Unable to participate" includes adult survivors of SAM who were unavailable because of migration (*n* = 53), illness (*n* = 19), refusal (*n* = 14), or pregnancy (*n* = 30). TMRU, Tropical Metabolism Research Unit; UHWI, University Hospital of the West Indies; JAMAKAS, Jamaica Marasmus and Kwashiorkor Adult Survivors.

**Table 1. Measures of weight and height gain during rehabilitation and post-recovery from severe acute malnutrition (SAM).**

| | Weight & Height Gain Measurement | Definition | Time Frame |
|---|---|---|---|
| Rehabilitation Phase | Δ WAZ/day | Average change in WAZ per day | Date of minimum weight to date of maximum weight during hospitalization |
| | Δ g/kg/day | Average change in grams per kilogram of body weight per day | |
| | Δ g/day | Average change in grams per day | |
| Post-recovery Phase | Δ WAZ/month | Average change in WAZ per month | Date from hospital discharge to one year post-recovery |
| | Δ g/kg/month | Average change in grams per kilogram of body weight per month | |
| | Δ HAZ/month | Average change in HAZ per month | |

WAZ- weight-for-age z-scores, HAZ-height-for-age z-scores

not SAM. Due to individual variations in the attainment of maximum weight, weight gain was expressed as 'per day'.

The remaining three measures represented weight gain and height gain from the time of hospital discharge until one year post-discharge. For these measures, WAZ, g/kg and HAZ were expressed as rate of change per month (**Table 1**).

Outcome variables included BMI, waist circumference, fat mass, fat mass index, % fat mass, android fat, android: gynoid (AG) ratio, lean body mass, lean mass index, systolic blood pressure, diastolic blood pressure, fasting glucose, fasting insulin, LDL-cholesterol, triglyceride, and homeostatic model assessment for insulin resistance (HOMA-IR). Body composition (height-adjusted lean mass and fat mass) was assessed by dual-energy X-ray absorptiometry (DEXA).

Lean mass index based on DEXA was calculated using the formula: lean mass/height$^2$, and similarly fat mass index based on DEXA was calculated using the formula: fat mass/height$^2$, each in BMI units of kg/m$^2$. HOMA-IR was calculated using the formula:

$$\text{HOMA} - \text{IR} = \text{fasting insulin}(\text{microU/L}) \times \text{fasting glucose}(\text{mmol/L})/22.5.$$

Statistical Analyses: Variables were tested for normal distribution using the Shapiro-Wilk test. To determine the associations between rehabilitation and post-recovery weight gain and post-recovery height gain (determinants) and adult body size, body composition and NCD risk factors (outcomes), we conducted separate linear regression analyses for each outcome adjusting for age and sex and (as an additional model) minimum WAZ (i.e., WAZ at the time of minimum weight). All outcome measures of body composition and blood pressure were adjusted for adult height. Adult BMI was treated as an outcome variable and not adjusted for in the analyses as it was considered to be on the causal pathway between growth and the outcomes. Each weight or height gain measure was evaluated as a continuous variable, and we reported differences in continuous outcomes per unit weight or height gain depending on the assigned measure. Anthropometric measurements were also grouped into quintiles which were used as ordinal variables in age- and sex-adjusted regression analyses. Quintiles 1 (Q1) and 5 (Q5) represented the slowest and fastest rates of weight and height gain respectively, and age- and sex-adjusted regression analyses were conducted comparing NCD risk across the weight and height gain quintiles. Finally, a latent class analysis (LCA) was conducted to qualitatively identify different weight and height gain pattern subgroups of the study population across 5 time points, viz., admission, time at minimum weight, time at maximum weight, discharge from hospital and one year post-recovery. The LCA was fitted using a generalised structural equation model with Gaussian-distributed outcomes at each timepoint included in the analysis. Model fit was assessed using Akaike Information Criteria (AIC); latent classes were extracted from the model with the lowest AIC, and which yielded reasonably sized classes. These LCA-predicted classes were plotted against the outcomes to facilitate interpretation. Stata version 16.0 (StataCorp LLC, College Station, Texas, USA) was used to conduct the statistical analyses and $p$-values $\leq 0.05$ were considered statistically significant.

## Results

There were 1,366 children admitted, of whom 47 died during treatment for malnutrition and the remaining 1,289 were discharged alive. The JAMAKAS study traced 729 adult survivors and enrolled 316 of them to the study. Subsequently, the CHANGE study identified 278 adult participants for whom childhood admission records were available for inclusion to this study. The recruitment of the adult survivors of SAM is summarised in **Fig 1**.

We analysed data for 278 adults (60% male) who were treated for severe malnutrition as children. Mean age at admission was 10.9 months, 88% were stunted (HAZ < -2), 77% and 87% were wasted (WHZ < -2 and MUAC < 12.5 cm respectively) and 65% had oedematous malnutrition. As adults, median age (IQR) was 26.5 (11) years, and median BMI (IQR) was 22.5 (6.1) kg/m$^2$ (**Table 2**).

Subsequent analyses were based on 273 participants during rehabilitation and 84 participants post-recovery as some participants had missing maximum weight or discharge weight data. In 273 participants, mean weight gain during rehabilitation was, according to the three measures, 0.07 WAZ/day, 10.1 g/kg/day and 62 g/day (**Table 3**). Mean (SD) change in WAZ during rehabilitation was 2.2 (0.97). In the post-recovery phase, weight gain was much slower: mean (SD) weight gain was 0.8 (0.5) g/kg/day and mean (SD) change in WAZ/ day was 0.002 (0.004) for 84 participants. Each of the six measures of weight and height gain was grouped in quintiles (**Table 3** and **Fig 2**).

## Associations between rehabilitation weight gain and adult NCD risk

The three measures of rehabilitation weight gain (n = 273) were associated with adult NCD risk in regression models adjusted for age, sex and minimum WAZ as follows:

Rehabilitation weight gain as ΔWAZ/day was positively associated with both waist circumference (adjusted regression coefficient = 53, 95% CI: 6, 100, $p$ = 0.03) and lean mass (adjusted regression coefficient = 37.0, 95% CI: 7, 67, $p$ = 0.02) (**S1 Table**). Those with the fastest weight gain had higher BMI (difference = 0.4 kg/m$^2$, 95% CI: 0.02, 0.8, $p$ = 0.04), waist circumference (difference = 1.2 cm, 95% CI: 0.3, 2, $p$ = 0.01), lean mass (difference = 0.8 kg, 95% CI: 0.2, 1.4, $p$ = 0.01), fat mass (difference = 0.8 kg, 95% CI: 0.01, 1.7, $p$ = 0.048) and % fat mass (difference = 0.85, 95% CI: 0.02, 1.7, $p$ = 0.04) than those with the slowest weight gain (**S2 Table** and **Fig 3**). Associations with waist circumference and % fat mass remained significant after further adjusting for adult height. Visual examination of the box plots (**Fig 3A, 3C, 3K and 3N**) suggested that only ΔWAZ ≥ 0.09/day was associated with measures of adult adiposity and when further exploratory analyses (excluding Q5) were conducted, ΔWAZ < 0.09/day was not associated with adult adiposity ($p$-values > 0.2) (data not shown). In the fastest growing quintile (Q5) in comparison to the slowest quintile (Q1), ΔWAZ/day was positively associated with BMI (difference = 2.1 kg/m2, 95% CI: 0.3, 3.8, $p$ = 0.02), waist circumference (difference = 6.5 cm, 95% CI: 2, 11, $p$ = 0.002), waist-hip ratio (difference = 0.03, 95% CI: 0.003, 0.05, $p$ = 0.03), fat mass (difference = 4.0 kg, 95% CI: 0.2, 8, $p$ = 0.04) and android fat mass (difference = 0.36 kg, 95% CI: 0.01, 0.7, $p$ = 0.05) (data not shown).

Rehabilitation weight gain in g/kg/day was positively associated with waist circumference (difference = 0.43 cm, 95% CI: 0.1, 0.8, $p$ = 0.02) and lean mass (difference = 0.28 kg, 95% CI: 0.05, 0.5, $p$ = 0.02) (**S1 Table**). Those with the fastest weight gain had higher BMI (difference = 0.5 kg/m$^2$, 95% CI: 0.1, 0.9, $p$ = 0.02), waist circumference (difference = 1.4 cm, 95% CI: 0.4, 2.4, $p$ = 0.005), fat mass (difference = 1.1 kg, 95% CI: 0.2, 2, $p$ = 0.02), fat mass index (difference = 0.32 kg/m2, 95% CI: -0.0001, 0.6, $p$ = 0.05), % fat mass (difference = 1.1, 95% CI: 0.2, 2, $p$ = 0.02), android fat mass (difference = 0.09 kg, 95% CI: 0.01, 0.2, $p$ = 0.03) and % android fat mass (difference = 1.2, 95% CI: 0.1, 2, $p$ = 0.03) than those with the slowest weight gain (**S2 Table** and **Fig 3**). The associations with waist circumference, % fat mass, android fat mass and % android fat mass remained significant after further adjusting for height (data not shown). Examination of the box plots (**Fig 3B, 3D, 3L, 3M and 3O-3Q**) suggested that only weight gain ≥ 12.9 g/kg/day was associated with measures of adult adiposity and when further exploratory analyses (excluding Q5) were conducted, rehabilitation weight gain <12.9 g/kg/day was not associated with adult adiposity ($p$-values > 0.3) (data not shown).

**Table 2. Clinical characteristics of 278 survivors of severe acute malnutrition.**

| Life Stage of the Study Population | Characteristics | At Admission N = 278 | At Discharge n = 276 | At 1 year PR n = 85 |
|---|---|---|---|---|
| Children | Age (months)** | 11.0 (0.33–36) | 12.7 (3–42) | 25.3 (15–45) |
| | % Male | 60.1% | 60.1% | 56.5% |
| | % Oedema | 65.5% | 0% | 0% |
| | Weight (kg) | 5.4 (1.5), 278 | 7.3 (1.8), 276 | 9.8 (1.6), 85 |
| | Height (cm) | 64.6 (7.0), 270 | 66.5 (7.2), 195 | 80.1 (5.6), 81 |
| | MUAC (cm) | 10.5 (2.0)*, 214 | 12.9 (1.2), 33 | 14.(1.5), 18 |
| | % Underweight (WAZ<-2) | 92 | 63 | 41 |
| | % Stunted (HAZ<-2) | 88 | 88 | 59 |
| | % Wasted (MUAC < 12.5cm) | 87 | 27 | 17 |
| | % Wasted (WHZ < -2) | 77 | 5 | 15 |
| | Days since admission | 0 | 49 (13–251)** | 406 (343–588)** |
| | **Characteristics** | **All N = 278** | **Male N = 167** | **Female N = 111** |
| Adults | Years since discharge | 27.2 (16–51) | 27.3 (16–51) | 26.5 (17–46) |
| | Age (years)* | 26.5 (11.3) | 27.0 (11.6) | 26.0 (11.8) |
| | Weight (kg)* | 64.4 (17.3) | 64.2 (12.4) | 65.2 (28.4) |
| | Height (cm) | 167.5 (9.0) | 171.1 (7.6) | 162 (7.8) |
| | BMI (kg/m$^2$)* | 22.5 (6.1) | 21.8 (4.0) | 24.8 (9.4) |
| | Waist circumference (cm)* | 75.6 (14.2) | 74.4 (9.9) | 80.2 (22.3) |
| | Waist-hip ratio * | 0.82 (0.07) | 0.82 (.06) | 0.81 (0.09) |
| | Diastolic blood pressure (mmHg)* | 71 (15) | 72 (14) | 70 (15) |
| | Systolic blood pressure (mmHg)* | 112 (14) | 115 (13) | 108 (13) |
| | Lean mass (kg)* | 49.4 (16.8) | 54.8 (8.6) | 39.2 (7.5) |
| | Lean mass index (kg/m$^2$) | 17.1 (2.6) | 18.5 (2.0) | 14.9 (1.8) |
| | Fat mass (kg)* | 9.7 (17.8) | 5.1 (6.3) | 23.5 (19.4) |
| | Fat mass index (kg/m$^2$)* | 3.2 (6.7) | 1.7 (2.2) | 8.6 (7.5) |
| | % Fat* | 15.5 (27.1) | 8.4 (8.5) | 36.8 (18.7) |
| | Android fat mass (kg)* | 0.66 (1.4) | 0.36 (0.50) | 1.6 (1.8) |
| | Android fat %* | 17.7 (31.9) | 10.4 (11) | 41.8 (23) |
| | Android-gynoid fat ratio* | 0.89 (0.30) | 0.90 (0.28) | 0.89 (0.31) |
| | Fasting blood glucose (mmol/L)*‡ | 4.6 (0.5) | 4.6 (0.5) | 4.5 (0.5) |
| | Fasting serum insulin (uIU/mL)*‡† | 3.6 (3.8) | 2.5 (2.3) | 5.6 (4.0) |
| | HOMA-IR*† | 0.71 (0.77) | 0.55 (0.45) | 1.0 (0.94) |
| | LDL-cholesterol (mmol/L)*† | 2.0 (0.85) | 1.9 (0.78) | 2.2 (1.3) |
| | Triglyceride (mmol/L)*† | 0.66 (0.38) | 0.68 (0.37) | 0.60 (0.44) |

N = 278. PR = post-recovery. WAZ = weight for age z-scores, HAZ = height for age z-scores, WHZ = weight for height z-scores. Z-scores based on WHO 2006 and 2007 references. MUAC = Mid upper arm circumference. Data are presented as means (± SDs), range** and medians (IQRs)*. "‡,‡†,†" n = 95, 83 and 82 respectively.

Rehabilitation weight gain in g/day was positively associated with lean mass (difference = 0.04 kg, 95% CI: 0.01, 0.1, $p = 0.02$) (**S1 Table**). Those with the fastest weight gain had greater waist circumference (difference = 1.2 cm, 95% CI: 0.2, 2, $p = 0.01$) and lean mass (difference = 0.7 kg, 95% CI: 0.1, 1.3, $p = 0.02$) than those with the slowest weight gain (**S2 Table and Fig 3**). Examination of the box plots (**Fig 3E**) suggested that rehabilitation weight gain $\geq 81$ g/day was associated with adult waist circumference and when further exploratory analyses were conducted, weight gain $< 81$ g/day was not associated with adult waist circumference ($p$-values $> 0.3$).

**Table 3. Descriptive data for quintiles of rehabilitation weight gain and post-recovery weight and height gain.**

| | Measures of Weight & Height Gain | Mean (SD) | Range |
|---|---|---|---|
| **1** | **Rehabilitation** | **0.07 (0.03)** | **-0.0025, 0.20** |
| | **Δ WAZ/ day (n = 273)** | | |
| | Q1 | 0.03 (0.01) | -0.00, 0.04 |
| | Q2 | 0.05 (0.00) | 0.04, 0.06 |
| | Q3 | 0.06 (0.00) | 0.06, 0.07 |
| | Q4 | 0.08 (0.00) | 0.07, 0.09 |
| | Q5 | 0.11 (0.02) | 0.09, 0.20 |
| **2** | **Rehabilitation** | **10.1 (3.8)** | **0.32, 27** |
| | **Δ g/kg/day (n = 273)** | | |
| | Q1 | 5.5 (1.3) | 0.3, 7.0 |
| | Q2 | 7.9 (0.6) | 7.0, 8.9 |
| | Q3 | 9.7 (0.5) | 8.9, 10.7 |
| | Q4 | 11.8 (0.6) | 10.7, 12.9 |
| | Q5 | 15.7 (2.9) | 12.9, 27.4 |
| **3** | **Rehabilitation** | **61.6 (25.3)** | **2.5, 189** |
| | **Δ g/day(n = 273)** | | |
| | Q1 | 31.8 (8.0) | 2.5, 41.0 |
| | Q2 | 46.9 (3.1) | 41.4, 51.7 |
| | Q3 | 58.6 (3.9) | 52.0, 64.8 |
| | Q4 | 71.6 (5.1) | 65.0, 81.0 |
| | Q5 | 99.9 (20.5) | 81.1, 188.7 |
| **4** | **Post-recovery** | **0.05 (0.1)** | **-0.22, 0.46** |
| | **Δ WAZ/ month(n = 84)** | | |
| | Q1 | -0.09 (0.06) | -0.22, -0.03 |
| | Q2 | -0.01 (0.01) | -0.03, 0.01 |
| | Q3 | 0.05 (0.02) | 0.01, 0.08 |
| | Q4 | 0.11 (0.02) | 0.08, 0.15 |
| | Q5 | 0.21 (0.08) | 0.15, 0.46 |
| **5** | **Post-recovery** | **24.9 (15.6)** | **-0.5, 7.1** |
| | **Δ g/kg/month (n = 84)** | | |
| | Q1 | 5.4 (8.0) | -12.9, 14.4 |
| | Q2 | 16.9 (2.4) | 14.6, 20.9 |
| | Q3 | 24.5 (2.2) | 21.4, 28.0 |
| | Q4 | 32.0 (3.3) | 28.1, 37.2 |
| | Q5 | 47.2 (12.9) | 37.8, 90.2 |
| **6** | **Post-recovery** | **0.10 (0.12)** | **-0.08, 0.54** |
| | **Δ HAZ/ month (n = 47)** | | |
| | Q1 | -0.04 (0.03) | -0.08, 0.01 |
| | Q2 | 0.04 (0.01) | 0.02, 0.06 |
| | Q3 | 0.09 (0.01) | 0.07, 0.11 |
| | Q4 | 0.14 (0.02) | 0.12, 0.17 |
| | Q5 | 0.29 (0.11) | 0.18, 0.54 |

Q - quintile: 1- slowest growth to 5- fastest growth, WAZ - weight-for-age z-scores, HAZ - height-for-age z-scores

None of the 3 measures of rehabilitation weight gain was associated with adult blood pressure or blood glucose, insulin or lipids.

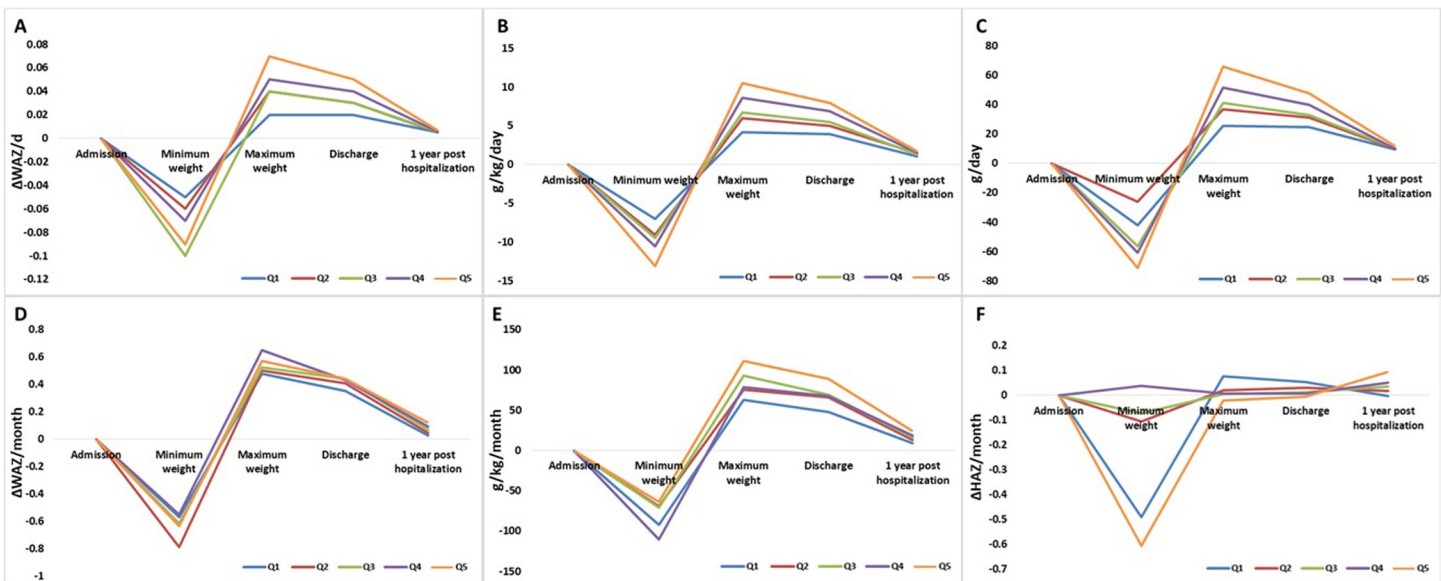

**Fig 2. Mean growth rates across two time periods according to the six weight and height gain measures.** A to C: (*rehabilitation weight gain*), D and E: (*post-recovery weight gain*) and F: (*post-recovery height gain*). Each measure was split into quintiles where Q1 = slowest gain, and Q5 = fastest gain. Values at admission acted as the comparator for the later timepoints.

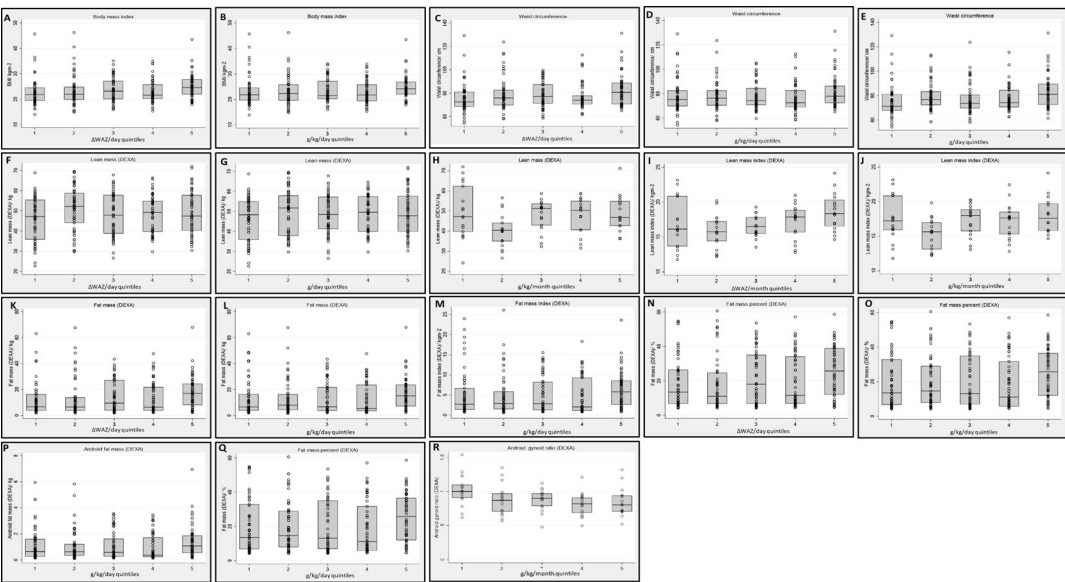

**Fig 3. Association of NCD outcomes with weight gain measures.** Age, sex and minimum weight WAZ-adjusted linear regression results (coefficient, *p*-value): **A to B** - body mass index: ΔWAZ/day (B = 0.4, *p* = 0.04), g/kg/day (B = 0.5, *p* = 0.02); **C to E** - waist circumference: ΔWAZ/day (B = 1.2, *p* = 0.01), g/kg/day (B = 1.4, *p* = 0.01), and g/day (B = 1.2, *p* = 0.01); **F to H** - lean mass: ΔWAZ/day (B = 0.8, *p* = 0.01), g/day (B = 0.7, *p* = 0.02) and g/kg/month (B = 1.3, *p* = 0.02); **I to J** - lean mass index: ΔWAZ/month (B = 0.35, *p* = 0.03) and g/kg/month (B = 0.43, *p* = 0.01); **K to L** - fat mass: ΔWAZ/day (B = 0.84, *p* = 0.048) and g/kg/day (B = 1.1, *p* = 0.01); **M** - fat mass index: g/kg/day (B = 0.32, *p* = 0.05); **N to O** - fat mass percent: ΔWAZ/day (B = 0.85, *p* = 0.04) and g/kg/day (B = 1.1, *p* = 0.02); **P** - android fat mass: g/kg/day (B = 0.09, *p* = 0.03); **Q** - android fat mass %: g/kg/day (B = 1.2, *p* = 0.03); **R** - android-gynoid ratio: g/kg/month (B = -0.03, *p* = 0.045). Quintile 1 Q1): slowest weight gain, Quintile 5 (Q5): fastest weight gain.

## Associations between post-recovery weight and height gain and adult NCD risk

Post-recovery weight gain (n = 84) and height gain (n = 47) were positively associated with NCD risk in regression models adjusted for age, sex and minimum WAZ as follows. Post-recovery weight gain as ΔWAZ/month was associated with lean mass index (difference = 4.6 kg/m$^2$, 95% CI: 0.6, 9, $p$ = 0.03) (S1 Table). Those with the fastest weight gain had a higher lean mass index (difference = 0.4 kg/m$^2$, 95% CI: 0.04, 0.7, $p$ = 0.03) than those with the slowest weight gain (S2 Table and Fig 3).

Post-recovery weight gain in g/kg/month was associated with lean mass index (difference = 0.03 kg/m$^2$, 95% CI: 0.01, 0.06, $p$ = 0.02) (S1 Table). Those with the fastest weight gain had higher lean mass (difference = 1.3 kg, 95% CI: 0.3, 2.4, $p$ = 0.02) and lean mass index (difference = 0.4 kg/m$^2$, 95% CI: 0.1, 0.7, $p$ = 0.01) and a lower android: gynoid ratio (difference = -0.03, 95% CI: -0.07, -0.001, $p$ = 0.05) than those with the slowest weight gain (S2 Table and Fig 3).

Post-recovery height gain as ΔHAZ/month was not associated with NCD outcomes and there was no difference between those who gained height fastest and those who gained height slowest.

None of the 3 measures of post-recovery weight or height gain was associated with adult blood pressure or blood glucose, insulin or lipids.

## Latent class analysis

Latent class analysis (LCA) identified a model with 6 interpretable classes of at least 21 children each which had the lowest Akaike Information Criteria (3159) (Fig 4). Class 1 (10%) consisted of the children who were most underweight on admission (mean WAZ < -6) and with the least weight gain during rehabilitation. Class 2 (21%) had the second most underweight children with the second lowest rehabilitation weight gain. Class 3 (30%) were children with the second highest rehabilitation weight gain. Class 4 (18%) had the highest rehabilitation weight gain. Class 5 (13%) had the second highest WAZ at admission and the third highest rehabilitation weight gain. Class 6 (8%) were the children who were least underweight on admission. All classes have a similarly steep level of catch-up during rehabilitation, except class 4. After discharge, class 1, with the greatest deficits, continued to catch up, and there was a pattern of decreasing catch-up weight and height gain with increasing classes, until class 6 where "catch down" was observed (Fig 4).

Age and sex-adjusted regression analyses against the NCD outcomes were carried out with LCA class as a categorical variable and Class 1 being the reference group. In children in Class 5 (of which 89% had oedematous malnutrition), change in WAZ was associated with greater adult fat mass (difference = 5.2kg, SE = 2.6; 95% CI: 0.1, 10, $p$ = 0.045) and android fat mass (difference = 0.52 kg, SE = 0.24; 95% CI: 0.04, 0.99, $p$ = 0.03) compared to children in Class 1.

## Discussion

Weight gain targets during treatment of severe malnutrition have been in place since the 1970s, with the recommendation that a desirable rehabilitation weight gain target is over 10g/kg/day [23]. This analysis of adult survivors of severe malnutrition in childhood demonstrated that, during the rehabilitation phase of malnutrition treatment, weight gain above certain thresholds is associated with risk factors for NCDs (i.e., greater BMI, waist circumference, fat mass and android fat mass). Weight gain during the first year post-recovery was inversely associated with android-gynoid fat ratio. Faster weight gain during and after admission for SAM was associated with greater lean mass in adult survivors, which is typically associated with lower NCD risk. It is

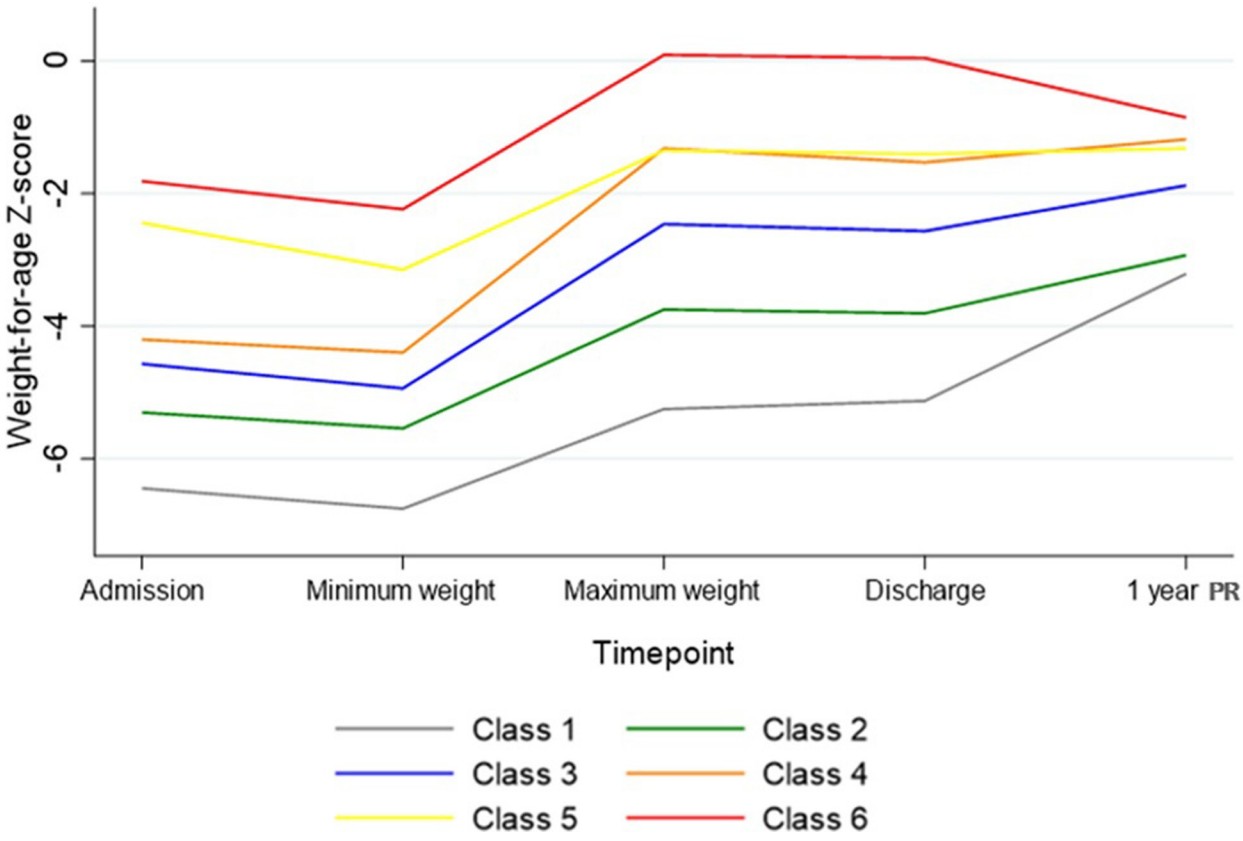

**Fig 4. Latent class analysis of patterns of weight-for-age z-score from admission to 1 year post-recovery for SAM in 278 adults.** Classes ranked by lowest WAZ upon admission: 1, 2, 3, 4, 5, 6, and by highest weight gain during rehabilitation: 4, 3, 6, 5, 2, 1. PR- post-recovery.

important to note that these associations are independent of weight-for-age at the time of hospitalization and adult height. Neither rehabilitation weight gain nor post-recovery weight gain was associated with adult blood pressure or blood glucose, insulin or lipids.

## Rehabilitation weight gain and adult adiposity

We have demonstrated that faster weight gain during malnutrition rehabilitation is associated with greater adult BMI; and higher BMI, being an indicator of excess or inadequate weight, is itself an independent risk factor for cardiovascular disease. Faster weight gain during nutritional rehabilitation is also associated with increased adult adiposity independent of adult height. Excess adipose tissue is known to contribute to the production of proinflammatory cytokines which are also associated with increased risk of cardiovascular disease. Importantly, faster rehabilitation weight gain was associated with fat mass index, which has been shown to be a more complete tool for diagnosing obesity than both BMI and fat mass as it differentiates fat and muscle mass of the total body weight [24]. Furthermore, fat accumulation in the android compartment is specifically known to be associated with increased cardiovascular and metabolic risk. Studies in younger populations have also demonstrated that android fat is more closely related to metabolic risk factors [25]. This tendency for adult SAM survivors to accumulate visceral fat may be a consequence of adipose tissue maldevelopment which includes altered body composition and increased visceral fat [26]. Rapid postnatal infant weight gain has been shown to be associated with increases in both visceral and abdominal

subcutaneous adipose tissue, as well as total adiposity and the risk of obesity in middle adulthood [27]. Additionally, increased visceral adipose tissue is linked to high triglycerides and low HDL cholesterol levels [28]. However, while the tendency to store fat abdominally might be a persisting response to adverse conditions and growth failure in fetal life and infancy [29], we have shown this tendency to store fat in the abdominal compartment to be independent of minimum weight-for age z-score during hospitalization. Similarly, we recently reported that children who experienced faster weight gain during nutritional rehabilitation had greater liver fat as adults, and this relationship was independent of weight-for age z-score at admission [30]. Therefore, the association between rehabilitation weight gain in childhood and adult android fat in this group of young, normal-weight SAM survivors could signal an increased risk of clinical NCDs as they age.

In this group of adult SAM survivors, measures of adiposity (i.e., BMI, waist circumference, fat mass and android fat mass) appear similar across the first 4 quintiles of rehabilitation weight gain (defined as ΔWAZ, g/kg/day and g/day), whereas the fifth quintile showed greater adult adiposity. Thus, each unit increase in weight gain does not appear to increase NCD risk in the same way, and there seems to be a threshold effect of rehabilitation weight gain on adult adiposity. This suggests that weight gain at or above specific thresholds (i.e., 13 g/kg/day, ΔWAZ > 0.09/day, 81g/day) could impute greater adult adiposity.

It is also important to point out that although our findings indicate a statistically significant association across the range of NCD risk factors, these individuals have not exceeded established thresholds for adverse outcomes such as obesity. So, while faster weight gain was associated with higher adult BMI, very few of these participants were obese (i.e., BMI >30 kg/m$^2$). Additionally, several studies have reported a lower susceptibility to visceral adipose tissue deposition in blacks compared to whites [31, 32]. Therefore, although it is unclear whether the levels of visceral fat in these study participants are sufficient to impute cardiometabolic risk, the possibility cannot be ruled out.

Our study produced the unexpected finding of an inverse association between post-recovery weight gain (g/kg/month) and android-gynoid percent fat ratio (AG), an important predictor of metabolic and cardiovascular disease risk. Indeed, in over 1,800 participants with a mean age of 35 years, AG was reported to have a greater association with cardiometabolic dysregulation than android fat, gynoid percent fat or BMI [33]. While our findings are counterintuitive, they further emphasize the complexity of these associations which may depend not only on the extent and rate of weight gain, but also the age of the child at the time of weight gain, and other factors outside the scope of this study.

### Rehabilitation and post-recovery weight gain and adult lean mass

Lean mass in adult SAM survivors was associated with both rehabilitation and post-recovery weight gain. While greater lean mass is generally associated with reduced NCD risk, the association of lean mass with health appears complex [34]. Although lean mass incorporates muscle mass, which is widely considered to protect against diabetes, high levels of lean mass have been associated with higher blood pressure [35, 36]. Additionally, it is possible that in this group, lean mass is accreted in tandem with fat mass, with the benefit of higher lean mass being obscured by the deleterious effects of fat (particularly visceral fat) accumulation.

### The role of minimum WAZ

It is important to establish the role of minimum weight-for-age z scores in these analyses as it has been reported that the most underweight children tend to gain more weight during nutritional rehabilitation. In our participants, however, most of the reported associations between

both rehabilitation and post-recovery weight gain and adult adiposity were independent of minimum WAZ. However, in one subgroup of participants identified by latent class analysis, minimum WAZ appears to have a discernible effect. Therefore, the relative importance of rehabilitation weight gain versus admission WAZ in predicting NCD risk in these adult SAM survivors may be dependent on how weight gain is defined, and there could be an interplay between these two factors influencing the association between childhood weight gain and adult adiposity.

## Rehabilitation and post-recovery weight gain

Rapid weight gain in infancy as a dichotomous variable has been defined as a weight-for-age Z score gain greater than 0.67 between birth and 1.5 years [37]. Others have defined it as a change in WAZ > 0.67 over an unspecified time period [38, 39]. Our study participants had a mean change in WAZ during rehabilitation (mean duration 37 days) that was more than 3 times the + 0.67 cut-off. In this study, ΔWAZ/day was associated with very large differences in waist circumference and lean mass, reflecting the fact that a 1 unit change in WAZ in a day is extremely high and very unlikely to occur. However, if we were to apply our mean ΔWAZ/day of 0.07, it would be associated with a difference in adult waist circumference of 3.7cm and adult lean mass of 2.6kg.

Our findings call current guidelines which recommend that a weight gain of >10 g/kg/day during rehabilitation into question, as some of our participants who gained more than 10/g/kg/day during rehabilitation were shown to have greater adiposity as adults. In India, WAZ and g/kg/day disagreed substantially as methods for measuring weight gain in more than a third of preterm children studied [40]. How the two measures agree in children born at term is unclear, however, among our participants the two measures were similarly associated with measures of adult adiposity but not lean mass.

In summary, while all three measures of rehabilitation weight gain were associated with at least one measure of adult adiposity; the same was not true of the measures of post-recovery weight gain. Although z-scores are considered the most appropriate descriptors of malnutrition, among our participants rehabilitation weight gain in g/kg/day had the most consistent associations with adult NCD risk factors and has the additional advantage of being more applicable in clinical settings. As a result of this inconsistency, we were unable to draw conclusions regarding the most appropriate proxy for post-malnutrition weight gain from this study.

## Strengths and limitations

The strengths of the study include the uniqueness of the cohort and the availability of detailed anthropometric and body composition measures in children and adults. This is also one of very few studies to follow survivors of malnutrition for such a long period. Similar analysis of a Malawian cohort of children 7 years post-SAM also suggested possible NCD risk factors in those who had the fastest post-malnutrition weight gain [41] but whether this is maintained longer term remains to be seen. Whilst the consistency of association adds weight to this being a potentially causal relationship, work in other cohorts in other settings is needed. One major advantage of the LION Jamaica cohort over the Malawian one is the low in-programme mortality (∼ 4%): this minimizes the problem of healthy survivor bias.

However, we also acknowledge limitations. Firstly, rehabilitation weight gain in terms of body composition (lean mass vs fat mass) might have influenced the outcome, but these were not measured at recovery. Additionally, other post-recovery factors occurring in childhood, including the home diet and intercurrent illnesses, could have influenced the observed associations. Data relating to adult dietary intake, physical activity, and co-morbid conditions, all of

which may confound the observed associations, were not available. The participants in this study were Afro-Caribbean, and the findings may be different in other settings and ethnic groups. Finally, participants in this study were treated in inpatient feeding programmes when rates of weight gain were particularly fast. Most current SAM treatment programmes focus on outpatient-based care and have overall slower rates of weight gain [42, 43]. Patient profile also varies: WHO Growth Standards have replaced NCHS growth references as admission criteria in today's programmes and larger numbers of overall less wasted children are admitted [44]. It is unknown whether associations seen in this historical data from Jamaican inpatients also applies to faster growing children in today's SAM treatment programmes.

## Conclusions

In this unique cohort of adult Afro-Caribbean SAM survivors, we demonstrated significant associations between rehabilitation and post-recovery weight gain and adult NCD risk that were independent of admission weight and adult height. While the implications of some of the findings are mixed, we have presented evidence of risk of increased BMI and adiposity, especially abdominal adiposity, in these adult SAM survivors whose rehabilitation weight gain exceeded certain thresholds. Our findings challenge the existing guidelines relating to weight gain targets during malnutrition treatment in childhood and underscore the need for further characterisation of optimal post-malnutrition weight gain in an interventional trial.

## Supporting information

**S1 Table. Results of ungrouped linear regressions of rehabilitation weight gain and post-recovery weight and height gain against NCD risk indicators in 273 adult survivors of severe acute malnutrition.**
(DOCX)

**S2 Table. Results of quintile grouped linear regressions of rehabilitation weight gain and post-recovery weight and height gain against NCD risk indicators in 273 adult survivors of severe acute malnutrition.**
(DOCX)

## Acknowledgments

We gratefully acknowledge the men and women who took part in the study. We also recognize support from the members of the CHANGE Study Collaborators' Group: Gerard Bryan Gonzales, Estelle McLean, Kelda Yeung, Amir Kirolos, Abena Amoah, Mia Crampin, Tsinuel Nigatu, Jonathan Swann, Kenneth Maleta, Carlos Grijalva-Eternod and Robert C. Stewart.

## Author Contributions

**Conceptualization:** Debbie S. Thompson, Kimberley McKenzie, Charles Opondo, Michael S. Boyne, Natasha Lelijveld, Tim J. Cole, Kenneth Anujuo, Mubarek Abera, Melkamu Berhane, Albert Koulman, Stephen A. Wootton, Marko Kerac, Asha Badaloo.

**Formal analysis:** Debbie S. Thompson, Kimberley McKenzie, Charles Opondo, Tim J. Cole.

**Funding acquisition:** Marko Kerac.

**Methodology:** Debbie S. Thompson, Kimberley McKenzie, Michael S. Boyne, Marko Kerac.

**Supervision:** Marko Kerac.

**Writing – original draft:** Debbie S. Thompson.

**Writing – review & editing:** Debbie S. Thompson, Kimberley McKenzie, Charles Opondo, Michael S. Boyne, Natasha Lelijveld, Jonathan C. Wells, Tim J. Cole, Kenneth Anujuo, Mubarek Abera, Melkamu Berhane, Albert Koulman, Stephen A. Wootton, Marko Kerac, Asha Badaloo.

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
