## [Decision Letter · Decision Letter 0]

30 Aug 2023

PGPH-D-23-01420

Faster rehabilitation weight gain during childhood is associated with risk of non-communicable disease in adult survivors of severe acute malnutrition

Dear Dr. Thompson,

Thank you for submitting your manuscript to PLOS Global Public Health. After careful consideration, we feel that it has merit but does not fully meet PLOS Global Public Health’s publication criteria as it currently stands. Therefore, we invite you to submit a revised version of the manuscript that addresses the points raised during the review process.

We look forward to receiving your revised manuscript.

Kind regards,

Aditi Apte, MD PhD

Academic Editor

Journal Requirements:

2. We noticed that you used "data not shown" in the manuscript. We do not allow these references, as the PLOS data access policy requires that all data be either published with the manuscript or made available in a publicly accessible database. Please amend the supplementary material to include the referenced data or remove the references.

3. Please provide separate figure files in .tif or .eps format only and remove any figures embedded in your manuscript file. Please also ensure all files are under our size limit of 10MB.

4. We have noticed that you have a list of Supporting Information legends in your manuscript. However, there are no corresponding files uploaded to the submission. Please upload them as separate files with the item type 'Supporting Information'. 

Additional Editor Comments (if provided):

Reviewers' comments:

Reviewer's Responses to Questions

**Comments to the Author**

1. Does this manuscript meet PLOS Global Public Health’s publication criteria? Is the manuscript technically sound, and do the data support the conclusions? The manuscript must describe methodologically and ethically rigorous research with conclusions that are appropriately drawn based on the data presented.

Reviewer #1: Yes

Reviewer #2: Yes

2. Has the statistical analysis been performed appropriately and rigorously?

Reviewer #1: No

Reviewer #2: Yes

3. Have the authors made all data underlying the findings in their manuscript fully available (please refer to the Data Availability Statement at the start of the manuscript PDF file)?

Reviewer #1: Yes

Reviewer #2: Yes

4. Is the manuscript presented in an intelligible fashion and written in standard English?

Reviewer #1: Yes

Reviewer #2: Yes

5. Review Comments to the Author

Reviewer #1: 1) Line 261, 262 need to mention the direction of the relationship would bring more clarity

2) Supplementary table 1 – marking/highlighting the statistically significant relationship would be helpful in all tables

3) Line 271 to 275 – specifying the direction of the association will bring clarity in understanding

4) In these participants the weight gain could be of 2 types; 1) weight gain due to increase in lean mass, 2) weight gain due increase in fat mass. This aspect needs to be looked into separately, as in, the participants who have gained weight beyond the threshold was there a difference in the composition of the weight gain (whether lean mass was gained or fatmass was gained) and if so were the relationship to NCD risk differ. These being complex interactions there is a need to look into these interactions in depth.

5) As already mentioned by the authors in the limitation post recovery factors are very important and would be confounding to all the findings, hence need to be looked into. The present findings do highlight the point that there is a need to be cautious about the rehabilitation and post recovery weight gain but many of the important confounding factors like diet, activity and environment have not been looked into which might affect the results differently.

Reviewer #2: The manuscript is well written but there are things that can be improved before acceptance. Please refer to the attached file for my suggested edits and their corresponding highlighted sections in the manuscript.

6. PLOS authors have the option to publish the peer review history of their article (what does this mean?). If published, this will include your full peer review and any attached files.

**Do you want your identity to be public for this peer review?** For information about this choice, including consent withdrawal, please see our Privacy Policy.

Reviewer #1: No

Reviewer #2: **Yes: **Dehao Chen

---

## [Decision Letter · Decision Letter 1]

16 Nov 2023

Faster rehabilitation weight gain during childhood is associated with risk of non-communicable disease in adult survivors of severe acute malnutrition

PGPH-D-23-01420R1

Dear Dr. Thompson,

We are pleased to inform you that your manuscript 'Faster rehabilitation weight gain during childhood is associated with risk of non-communicable disease in adult survivors of severe acute malnutrition' has been provisionally accepted for publication in PLOS Global Public Health.

Best regards,

Giridhara R Babu, MBBS, MPH, PhD

Academic Editor

Reviewer Comments (if any, and for reference):

Reviewer's Responses to Questions

**Comments to the Author**

1. If the authors have adequately addressed your comments raised in a previous round of review and you feel that this manuscript is now acceptable for publication, you may indicate that here to bypass the “Comments to the Author” section, enter your conflict of interest statement in the “Confidential to Editor” section, and submit your "Accept" recommendation.

Reviewer #1: All comments have been addressed

Reviewer #2: All comments have been addressed

2. Does this manuscript meet PLOS Global Public Health’s publication criteria? Is the manuscript technically sound, and do the data support the conclusions? The manuscript must describe methodologically and ethically rigorous research with conclusions that are appropriately drawn based on the data presented.

Reviewer #1: Yes

Reviewer #2: (No Response)

3. Has the statistical analysis been performed appropriately and rigorously?

Reviewer #1: Yes

Reviewer #2: (No Response)

4. Have the authors made all data underlying the findings in their manuscript fully available (please refer to the Data Availability Statement at the start of the manuscript PDF file)?

Reviewer #1: Yes

Reviewer #2: (No Response)

5. Is the manuscript presented in an intelligible fashion and written in standard English?

Reviewer #1: Yes

Reviewer #2: (No Response)

6. Review Comments to the Author

Reviewer #1: This topic is of importance in view of the growing incidence of non communicable disease hence publishing this data is important.

The authors have addressed all my concerns and hence the manuscript can be accepted for publication

Reviewer #2: (No Response)

7. PLOS authors have the option to publish the peer review history of their article (what does this mean?). If published, this will include your full peer review and any attached files.

**Do you want your identity to be public for this peer review?** For information about this choice, including consent withdrawal, please see our Privacy Policy.

Reviewer #1: No

Reviewer #2: **Yes: **Dehao Chen
